# Clinical Developments and Challenges in Treating FGFR2-Driven Gastric Cancer

**DOI:** 10.3390/biomedicines12051117

**Published:** 2024-05-17

**Authors:** David K. Lau, Jack P. Collin, John M. Mariadason

**Affiliations:** 1Olivia Newton-John Cancer Research Institute, Heidelberg, VIC 3084, Australia; jack.collin@onjcri.org.au; 2School of Cancer Medicine, La Trobe University, Heidelberg, VIC 3084, Australia; 3Department of Medical Oncology, Peter MacCallum Cancer Centre, Melbourne, VIC 3000, Australia; 4Department of Oncology, Monash Health, Clayton, VIC 3168, Australia

**Keywords:** gastric cancer, FGFR2, targeted therapies, monoclonal antibodies

## Abstract

Recent advances in the treatment of gastric cancer (GC) with chemotherapy, immunotherapy, anti-angiogenic therapy and targeted therapies have yielded some improvement in survival outcomes; however, metastatic GC remains a lethal malignancy and amongst the leading causes of cancer-related mortality worldwide. Importantly, the ongoing molecular characterisation of GCs continues to uncover potentially actionable molecular targets. Among these, aberrant FGFR2-driven signalling, predominantly arising from *FGFR2* amplification, occurs in approximately 3–11% of GCs. However, whilst several inhibitors of FGFR have been clinically tested to-date, there are currently no approved FGFR-directed therapies for GC. In this review, we summarise the significance of FGFR2 as an actionable therapeutic target in GC, examine the recent pre-clinical and clinical data supporting the use of small-molecule inhibitors, antibody-based therapies, as well as novel approaches such as proteolysis-targeting chimeras (PROTACs) for targeting FGFR2 in these tumours, and discuss the ongoing challenges and opportunities associated with their clinical development.

## 1. Gastric Cancer Incidence and Current Treatments

Gastric cancer (GC) is the fifth most commonly diagnosed malignancy and the third leading cause of cancer-related mortality worldwide [1]. In 2020, there were approximately 1.1 million new GC diagnoses, and the disease was responsible for ~769,000 deaths, with most newly diagnosed cases presenting with metastatic disease. The incidence of GC is two-fold higher in men than in women and particularly prevalent in eastern Asia and eastern Europe. Whilst the global incidence of GC has been decreasing, the incidence has been increasing amongst younger adults (age < 50 years) for reasons not fully understood [1].

### 1.1. Molecular Subsets of Gastric Cancer

The genomic characterisation of GCs has revealed that these tumours can be subdivided into four molecular subtypes: microsatellite instability (MSI), Epstein–Barr virus (EBV) positive, chromosomal instability (CIN) and genomically stable (GS) [2,3]. Whilst all subtypes occur throughout the stomach, the prevalence of CIN tumours is highest in the proximal stomach (cardia) and gastroesophageal junction (GEJ) (65%) and is less frequent in the distal stomach (referred to as the “CIN gradient”), while MSI and GS tumours are more common within the body, fundus and pylorus [2]. Several other means of subclassifying gastric cancers with prognostic and predictive implications have also been identified by integrating genetic changes with transcriptomic and epigenetic data [3,4,5,6], in what is a continually evolving landscape.

### 1.2. Current Treatments for Metastatic Gastric Cancer

In the metastatic setting, cytotoxic chemotherapy is the current standard of care treatment for GC, where patients treated with combination regimens of platinum and fluoropyrimidine-based treatments have a median overall survival of 10–13 months [7,8]. In 2021, the CHECKMATE 649 trial showed that the addition of the anti-PD1 antibody nivolumab to fluoropyrimidine and platinum-containing chemotherapy further improved the overall survival compared to chemotherapy alone (13.6 vs. 11.8 months, HR 0.70 95% CI 0.61–0.81), and was FDA-approved for all GC patients [8]. Microsatellite instability high (MSI-H) status occurs in ~3% of GCs and was found to be a sub-group with long term responses to anti-PD1 therapy in the CHECKMATE 649 trial (38.7 months vs. 12.3 months) [8].

Other molecular subsets include HER2-amplified/expressing tumours, which account for ~22% of all cases [9], for which treatment with the HER2-directed monoclonal antibody trastuzumab (in combination with chemotherapy), and more recently the antibody–drug conjugate fam-trastuzumab deruxtecan-nxki, is clinically approved [9,10]. Additional approved treatments include the anti-angiogenic therapy ramucirumab, which is approved for use as second-line therapy either in combination with paclitaxel or as monotherapy [11,12]. 

With regards to emerging treatment targets, Claudin 18.2 is overexpressed in ~42% of HER2-negative gastric cancers. Zolbetuximab, a monoclonal antibody targeting Claudin 18.2 has recently been shown to prolong progression-free survival when combined with first-line chemotherapy (10.61 versus 8.67 months, HR 0.75, 95% CI 0.60–0.94; *p* = 0.0053), and regulatory approval of zolbetuximab is currently pending [13].

Despite these treatment advances, there are several molecular subsets of GC for which there are currently no targeted treatment options, and where there is a clear need to develop new treatments in order to improve outcomes. 

Fibroblast growth factor receptor 2 (FGFR2) is an important oncogenic driver of a subset of GCs, and FGFR2 targeting represents a significant treatment opportunity for patients harboring these tumours. However, despite decades of therapeutic development, there are currently no FGFR2-targeted therapies approved for GC. Nevertheless, recent clinical trials have shown encouraging signs of clinical activity for FGFR-targeting small-molecule inhibitors in subsets of *FGFR2*-amplified cases (discussed below), while FGFR2-targeting biologics have also shown promising clinical activity and are currently being evaluated in phase III studies in combination with chemotherapy and immunotherapy. Excitingly, several new strategies for targeting FGFR2 are also beginning to emerge. Herein, we critically review the pre-clinical and clinical research undertaken to date to target FGFR2, and outline the challenges and emerging strategies for improved targeting of these tumours and advancing these agents into routine clinical use.

## 2. FGFR Signalling

Fibroblast growth factor receptors (FGFRs) are a family of five transmembrane receptor tyrosine kinases (FGFR1–5) [14]. FGFR1, FGFR2, FGFR3 and FGFR4 comprise of three extracellular immunoglobulin (Ig)-like domains (Ig1-IgIII), a hydrophobic transmembrane segment and a cytoplasmic split tyrosine kinase domain. Comparatively, FGFR5 (FGFRL1) lacks an intracellular tyrosine kinase domain and may have an inhibitory role in FGFR signalling by acting as a decoy receptor for fibroblast growth factor (FGF) ligands, sequestering them from binding to FGFRs 1–4 [15]. 

In addition to the FGFRs, there are 18 mammalian FGFs, which are the natural ligands of these receptors, which, upon binding to an FGFR monomer at the IgII and IgIII region cause receptor dimerization. FGFs can be divided into five paracrine (FGF1, 4, 7, 8 and 9) and one endocrine (FGF19) subfamilies and bind to either a single or multiple FGFRs [16]. The binding of paracrine FGFs requires the cofactor heparan sulfate proteoglycan (HSPG), while Klotho proteins (α-Klotho/β-Klotho) are the cofactors required for binding of the endocrine FGFs [17]. Ligand-induced FGFR dimerization results in the phosphorylation of tyrosine residues (e.g., Y463, Y583, Y585, Y653, Y654, Y730 and Y776) within the intracellular tyrosine kinase domain, which are highly conserved between mammalian FGFR1 and FGFR2. Phosphorylated Y653 and Y654 act as allosteric regulators of kinase activity, while Y463, Y583, Y585, Y730 and Y776 act as docking sites for Src-homology 2 (SH2) domain proteins, which consequently activate downstream signalling pathways [18,19]. For example, fibroblast receptor substrate 2α (FRS2α) attachment to a phosphotyrosine residue triggers the recruitment of son of sevenless (SOS) and growth factor receptor-bound protein 2 (GRB2), consequently activating the RAS/MAPK and the PI3K signalling pathways, respectively. Other signalling pathways activated via FGFRs include the PLCγ/DAG/PKC and JAK/STAT (Figure 1). 

FGFR signalling pathways are also negatively regulated by a number of proteins including the FGF synexpression protein Sef (similar expression to FGF), a transmembrane protein that directly interacts with FGFR, and Sprouty proteins, which attenuate FGFR signalling by inhibiting GRB2 [16]. 

Alternative splicing of FGFRs 1–3 adds an additional layer of regulation to FGFR signalling. Specifically, alternate splicing of exons 8 and 9 in FGFRs 1–3 gives rise to tissue-specific alternate IgIII extracellular domain binding regions (IIIb in epithelial tissues from exon 8, and IIIc in mesenchymal tissues from exon 9), which are essential in determining FGF binding specificities [20,21]. FGFR4 lacks this alternative exon and therefore does not produce splice variants [22]. The range of FGF ligands and FGFR receptors results in highly regulated signalling in different physiological contexts [21].

## 3. Aberrant FGFR Signalling in Gastric Cancer

Dysregulation of the FGF/FGFR signalling pathway in GCs arises primarily through *FGFR* gene amplification, *FGFR* gain-of-function mutations, gene rearrangements, and alternative splicing events, which alter ligand binding, or through a combination of these aberrations [23].

### 3.1. FGFR2 Amplifications, Fusions and Mutations in GC

*FGFR2* gene amplifications, which result in receptor overexpression and constitutive oncogenic signalling independent of ligand binding, are the most common form of genetically-driven FGFR2 dysregulation in GC, occurring in 4–7% of all patients [24,25,26,27,28,29]. This varies according to patient ethnicity, with a large international study of resected GCs that determined *FGFR2* amplification status by *FGFR2* FISH (*FGFR2*/*CEP10* ratio < 2), reporting a prevalence of 7.4%, 4.6% and 4.2%, amongst patients from the UK, China and Korea, respectively [27]. 

In the cancer genome atlas (TCGA) patient cohort, *FGFR2*-amplified GCs were most frequently associated with the CIN (8%) and GS (9%) molecular subtypes [3]. Furthermore, *FGFR2*-amplified GCs are more likely to be of high tumour grade [30], and of diffuse-type histology [27,31,32], and several studies have associated *FGFR2* amplification status with poorer survival outcomes. For example, in studies of resected GC cases across all stages, *FGFR2* amplification was more likely to be associated with lymph node and distant metastasis, and poorer overall survival [27,31,33]. A similar association has also been observed in patients with metastatic GC treated with chemotherapy, where *FGFR2* amplification status was associated with poorer overall survival [27,30,33]. 

*FGFR2* gene fusions, resulting from chromosomal translocations, occur in 0.5–3% of GCs, although *FGFR2* is the most frequently perturbed among FGFR family members [28,29,34,35]. *FGFR2* fusion partners reported to date in GC include *TACC2*, *INPP5F*, *WDR11* and *BTBD16*, and in most cases, these tumours harboured concurrent *FGFR2* gene amplification [28,34,35]. Notably, both the *FGFR2* amplification and fusion events observed in GC often result in truncation of exon 18 (E18), and loss of the C-terminus of the FGFR2 kinase domain. This region plays an important role in the negative regulation of FGFR2, which may explain how these alterations confer oncogenic activity [36,37]. 

Finally, somatic *FGFR2* mutations have been reported in 0.5–3% of GCs [29,34,38,39,40], including the known hotspot mutations C382R and N549K, which affect the transmembrane domain and tyrosine kinase domain of the receptor, respectively [29,38,39,40,41].

### 3.2. Alternate Splicing of FGFR2

As described above, alternate splicing of FGFR2 is an additional mechanism of FGFR2-driven signalling regulation, as the alternatively spliced FGFR2 isoforms have different ligand binding affinities. FGFR2-IIIb (FGFR2b) is the predominant FGFR2 isoform expressed in epithelial tissues and has strong affinity for FGFs 1, 3, 7, 10 and 22, while FGFR2-IIIc (FGFR2c) is predominantly expressed in mesenchymal tissues and has high affinity for FGFs 1–2, -4, -6, -8, -9 -17 and -18 [42]. 

Using FGFR2b isoform-specific antibodies, several studies have demonstrated that FGFR2b overexpression assessed by immunohistochemistry correlates with *FGFR2* amplification, indicating that it is the predominant isoform expressed in these tumours, and also suggesting that it could be used as a cost-effective screening test to detect *FGFR2* amplification [43,44]. As expected, FGFR2b overexpression assessed in this manner was also associated with poorer survival [43,44]. Using novel antibodies that specifically detect the FGFR2b and FGFR2c isoforms, a recent study further confirmed that epithelial-specific FGFR2b is the predominant isoform overexpressed in GCs (4.5% of cases) [42], but also identified a small subset of GCs (0.7%), which over-express FGFR2c [42]. These tumours all also overexpressed FGFR2b, albeit in different tumour cells. Notably, although sample sizes were small, dual FGFR2b/c expressing GCs had a significantly poorer outcome compared to GCs that only overexpressed FGFR2b, which may be due to the expanded repertoire of FGFs capable of activating FGFR2 in these tumours [22].

### 3.3. FGFR2 Overexpression

Other immunohistochemistry-based studies that have examined the overall frequency of FGFR2 expression or overexpression in GC relative to normal gastric mucosa, have reported expression ranges between ~30% [45,46,47,48] to as high as 61% of tumours [49]. This may reflect differences in the antibodies used, or the method of scoring FGFR2 staining intensity. Nevertheless, high FGFR2 expression assessed by these methods was reported to be associated with more aggressive clinical features including tumour depth, lymph node and distant metastases [49,50], and worse overall survival in patients with primary gastric [47] and GEJ cancers [49,50]. Conversely, the predictive capacity of FGFR2 expression on chemotherapy benefit remains uncertain, with a recent report demonstrating no relationship between FGFR2 expression and first-line cytotoxic chemotherapy response rates [51], whereas other studies suggest that high FGFR2 expression may predict resistance to anti-HER2 therapies [52]. Importantly, our understanding of the associations between different levels of FGFR2 expression and the response to FGFR-targeted therapies continues to evolve. The establishment of standardized staining and scoring methods, possibly informed by parallel genomic analyses, is required if FGFR2 immunostaining is to be incorporated into routine clinical use as a companion diagnostic for predicting response to FGFR2-targeted therapies (discussed further below). 

## 4. Therapeutic Targeting of FGFR2 Using Small-Molecule Inhibitors

### 4.1. Small-Molecule Multi-Kinase Inhibitors

Several small-molecule multi-kinase inhibitors have been developed over the past two decades, primarily as anti-angiogenic agents, which inhibit a range of kinases including VEGFR, PDGFR, FLT3, RET, KIT and BCR-ABL. These include foretinib [53], cediranib (AZD2171) [54], ponatinib [55], sorafenib [56], sunitinib [57], pazopanib [58], nintedanib [59], lenvatinib [60], sulfatinib [61], dovitinib [62], lucitanib [63], SOMCL-085 [64], derazantinib [65], ODM-203 [66] and regorafenib [67] (Table 1). Consistent with the similarity of the bi-lobed structure of the FGFR protein kinase domain to that of all other protein kinases, these compounds also have FGFR inhibitory activity [55,56,59,68,69], albeit at modestly higher concentrations to which they inhibit VEGFR and PDGFR.

Consistent with their capacity to inhibit FGFR activity, several of these compounds, including cedirinib [69], pazopanib [58], ponatinib [68], dovitinib [55], sunitinib [57], SOMCL-085 [64], derazantinib [65], ODM-203 [66] and regorafenib [67] have been shown to effectively inhibit FGFR signalling, and to preferentially inhibit the proliferation of FGFR-altered GC cell lines over non-altered lines at clinically relevant concentrations [70,71,72]. 

However, whilst some of these inhibitors have demonstrated promising clinical activity in GC, there is currently insufficient evidence to suggest whether FGFR2 inhibition also contributes to their clinical efficacy. For instance, in the phase II INTEGRATE trial, which randomised 152 patients with treatment refractory metastatic GC to receive regorafenib or placebo [73], the overexpression of FGFRs 1–4 assessed by IHC was not associated with clinical benefit or objective response [74]. Similarly, in a single arm phase II study of nintedanib in patients with refractory oesophago-gastric cancers, 6 of 32 patients (19%) were progression-free at 6 months, however, FGFR2 alterations (which were detected in 18% of patients), were not predictive of progression-free survival (PFS) [75]. By contrast, FGFR2-expressing GC had a higher response rate and longer survival from combination treatment with pazopanib plus chemotherapy. Similarly, in a single arm phase II trial, patients with tumours overexpressing FGFR2 assessed by immunohistochemistry (13%) had a prolonged median PFS (8.5 vs. 5.6 months; *p* = 0.050), a higher response rate (87% vs. 69.5%), and a trend towards a prolonged overall survival (13.2 vs. 11.4 months; *p* = 0.055) [76]. Finally, the GASDOVI-1 clinical trial of dovitinib in *FGFR2*-amplified GC was completed several years ago, however, the results have not yet been reported (NCT01719549). Additional studies are therefore still needed to establish whether *FGFR2*-amplified/overexpressing cases derive greater benefit from multi-kinase inhibitors.

### 4.2. Small-Molecule Pan-FGFR Inhibitors

Several orally bioavailable pan-FGFR small-molecule inhibitors with specificity for the FGFR 1–4 kinase domain have also been developed. These include non-covalent pan-FGFR inhibitors (e.g., erdafitinib/JNJ-42756493, infigratinib/BGJ398, pemigatinib/INCB054828, AZD4547, CH5183284, LY2874455, E7090), irreversible covalent FGFR inhibitors (e.g., FIIN-1, PRN-1371, futibatinib/TAS-120), FGFR extracellular domain allosteric inhibitors (e.g., alofanib) and more recently, FGFR2-selective inhibitors (e.g., RLY-4008, further discussed below). Of these, only erdafitinib, futibatinib and pemigatinib are FDA-approved for the treatment of FGFR-altered bladder cancer and cholangiocarcinoma [77,78], with none approved for the treatment of GC. 

Nevertheless, several of these compounds have been shown to inhibit FGFR signalling and to inhibit the growth of FGFR-altered GC cell lines in vitro and in vivo in pre-clinical studies, including AZD4547 [79], infigratinib [67,80], erdafitinib [67,81], futibatinib [67,82], pemigatinib [83], LY2874455 [84] and E7090 [85]. Unsurprisingly, early phase trials of these compounds in unselected GC patients demonstrated only limited clinical benefit. For example, in a phase I study of the oral pan-FGFR inhibitor LY2874455, of 15 GC patients evaluated for efficacy, only one patient had a partial response, and the patient’s tumour was not *FGFR2* amplified [86]. Similarly, in a phase Ib study of the FGFR extracellular allosteric inhibitor alofanib in 21 heavily pre-treated molecularly unselected GC patients, there was only one partial response [87]. 

However, subsequent trials in selected patients with *FGFR* gene aberrations have also been largely disappointing (further discussed in Section 6). For example, in a phase I study of futibatinib, two partial responses were observed in a cohort of 9 patients with GC, only one of whom had an *FGFR2* amplification and the other a *FGFR3*–*TACC3* fusion [88]. The randomised phase II SHINE trial also failed to report a clinical benefit for treating GC patients with *FGFR2* amplification or polysomy with the FGFR inhibitor AZD4547. The study screened 960 patients with GC for *FGFR2* amplification or polysomy, and eligible patients (*n* = 71) were randomised to receive AZD4547 (amplified *n* = 18, polysomy *n* = 20) or paclitaxel chemotherapy (amplified *n* = 15, polysomy *n* = 15, *n* = 30). The trial failed to meet its primary endpoint of PFS, with a median PFS of 1.8 months and 3.5 months in the AZD4547 and paclitaxel arms, respectively (HR 1.31, 80% CI 0.89–1.95). The objective response rate (ORR) was 2.6% with AZD4547 and 23.3% in the paclitaxel arm. The lack of enrichment of high-level *FGFR2*-amplified tumours in this study may have contributed to the lack of response to AZD4547 [89]. 

Nevertheless, several trials involving multi-kinase or pan-FGFR kinase inhibitors are currently ongoing in patients with FGFR2-altered tumours including GCs. These include the multi-kinase inhibitor derazantinib, which is currently being tested as monotherapy as well as in combination with anti-PD-1 or chemotherapy in FGFR2-altered GC (NCT04604132), and anlotinib, which is currently being tested in combination with chemotherapy in patients with metastatic GC (ChiCTR1900026291). Trials of the irreversible covalent binding FGFR inhibitors futibatinib (TAS-120, NCT04189445) and infigratinib (NCT05019794) for FGFR 1–4-rearranged solid tumours and *FGFR2*-amplified GC are also currently ongoing (Table 2).

### 4.3. FGFR2-Specific Inhibitors

An exciting recent study from Relay Therapeutics described the development of lirafugratinib (RLY-4008), a highly selective, irreversible FGFR2 kinase domain inhibitor with >250- and >5000-fold selectivity for FGFR2 over FGFR1 and FGFR4 [90,91,92], developed by exploiting differences in the conformational dynamics between FGFR2 and other FGFRs. In pre-clinical studies, lirafugratinib inhibited FGFR2 phosphorylation, downstream signalling and proliferation of *FGFR2*-amplified SNU-16 GC cells in vitro and in vivo, and notably induced significantly less hyperphosphatemia compared to pan-FGFR inhibitors [90]. Intriguingly, lirafugratinib was also able to suppress signalling and tumour growth induced by common FGFR2 kinase domain mutations associated with acquired resistance to pan-FGFR inhibitor treatment in cholangiocarcinoma patients such as the *FGFR2^V564F^* gatekeeper mutation. 

Lirafugratinib is currently being clinically tested in a phase I/II tumour agnostic study of patients with *FGFR2* alterations (ReFocus trial), comprising a dose escalation, dose expansion and extension phase (NCT04526106) (Table 2). Recent data presented in abstract form [93] from the Phase I/2 ReFocus trial reported on the efficacy in 98 patients with FGFR2-altered solid tumours including GC patients (*n* = 22). Responses were observed in eight tumour types including GC where the ORR was 18% and the disease control rate (DCR) was 64%. The drug was reported to have a manageable toxicity profile including fewer off-isoform effects including hyperphosphatemia (discussed below). 

## 5. Factors Limiting Response to Small-Molecule FGFR2-Targeted Therapies in GC

Inherent, adaptive and acquired resistance to small-molecule FGFR inhibitors is currently the major challenge limiting the clinical progress of these agents, and which needs to be overcome to enable the clinical progression of these treatments. In addition, toxicities associated with the use of small-molecule FGFR inhibitors represent a further clinical challenge. The various modes of resistance described to date and the toxicities associated with small-molecule FGFR inhibitors are discussed below. 

### 5.1. Tumour Expression of FGFR2

One emergent theme from the clinical studies undertaken to date is that homogeneous, high-level *FGFR2* amplification may be predictive of the response of GCs to small-molecule FGFR inhibitors. For example, in the phase I study of the pan-FGFR inhibitor E7090 in patients with advanced solid tumours, the sole partial response observed was in a GC patient harboring a high-level *FGFR2* amplification (copy number 51) [93]. Similarly, in a single-institution translational study of 9 patients with *FGFR2*-amplified GC treated with the pan-FGFR inhibitor AZD4547, of the 3 patients who achieved an objective response (33.3%), all had homogenous, high-level *FGFR2* amplification [94]. 

These findings suggest that the criteria used to assess likelihood of response to small molecule FGFR2 inhibitors needs to be further developed, and assessment of FGFR2 protein overexpression by IHC alone may not be sufficient. This is further underscored by the extensive intratumoural heterogeneity in FGFR2b protein expression in GCs (55.5% of cases) [44], discordance in FGFR2 protein expression between primary and matched metastatic lymph node samples (28% of cases) [44], and the high false negative rates of detecting FGFR2 overexpression in a single tumour biopsy (up to 40% of cases), requiring increasing the number of diagnostic biopsies [95]. 

One possible strategy for more successfully predicting the response to FGFR2 inhibitors could be the use of circulating tumour DNA (ctDNA) technology [96]. For instance, the Japanese GI-SCREEN and GOZILA molecular profiling studies revealed that *FGFR2* amplification in patients with advanced GC was more frequently detected by ctDNA sequencing (7.7% of cases) compared to by tissue analysis alone (2.6–4.4% of cases), and notably 2 patients with *FGFR2* amplification detected by ctDNA sequencing after tumour progression but not by tissue analysis of the pre-treatment sample had responses to FGFR inhibitors [97].

### 5.2. Co-Occurrence of Other Oncogenic Drivers

Genomic profiling studies have revealed that approximately 20% of GCs harbouring *FGFR2* alterations (amplifications, mutations, fusions) also contained genetic alterations in either *MYC* (17%), *KRAS* (10%), *HER2* (10%), *EGFR* (8%), *PI3K* (6%) or *MET* (3%), which could potentially impact the sensitivity to FGFR inhibition [37,98]. Notably, synergistic anti-tumour activity was observed in an *FGFR2*-amplified and MET overexpressing GC patient-derived xenograft (PDX) model treated with the FGFR inhibitor AZD4547 and the MET inhibitor crizotinib, confirming the capacity of *MET* to confer at least partial resistance to FGFR inhibition [99]. 

A kinome-wide CRISPR/Cas9 screen of *FGFR2*-amplified SNU-16 and Kato III cells also identified a number of additional regulators of sensitivity to FGFR inhibition including kinases involved in the ILK, SRC and EGFR signalling [100]. Providing functional validation of these targets, combination treatment of an FGFR inhibitor and either an EGFR inhibitor (lapatinib) or an ILK inhibitor (Cpd22) synergistically inhibited the proliferation of FGFR2-altered GC cell lines [100]; however, whether these factors drive resistance in a clinical context remains to be demonstrated. 

Finally, mutations in *TP53* have been associated with resistance to FGFR inhibitors in several tumour types [101] but have not yet been carefully examined in GC. Given *TP53* mutations are a common event in GC, including ~50% of *FGFR2*-amplified cases, this would be an important biomarker to assess in ongoing clinical trials [101]. 

### 5.3. Adaptive Resistance to Small-Molecule FGFR Inhibition

The capacity of *FGFR2*-amplified GC lines to rapidly develop resistance to FGFR inhibition has also been demonstrated in pre-clinical models. For example, research from our own laboratory demonstrated that MAPK signalling (pERK1/2) is reactivated within 24–72 h of FGFR inhibitor treatment of *FGFR2*-amplified GC cell lines in vitro [74]. Importantly, this could be overcome by combination treatment with the clinically approved MEK inhibitor trametinib, revealing a potential strategy for enhancing the efficacy of these treatments [74]. 

### 5.4. Acquired Resistance to Small-Molecule FGFR Inhibition in GC

Several studies have also generated models of acquired resistance to FGFR inhibitors by continuous culture of *FGFR2*-amplified GC cell lines in the presence of FGFR inhibitors. For example, Grygielewicz et al. [102] generated FGFR-inhibitor-resistant derivatives of *FGFR2*-amplified SNU-16 cells by continuously culturing cells with increasing concentrations of the FGFR inhibitors AZD4547, BGJ398 or PD173074 for 1 month. Molecular characterization of the resistant line revealed a robust induction of an epithelial-to-mesenchymal transition (EMT), evidenced by increased vimentin and decreased E-Cadherin expression, and upregulation of multiple signalling pathways, including ERK/MAPK, STAT3, TGFb/SMAD and PI3K/AKT [102]. Lee et al. confirmed the induction of EMT in a follow-up study by generating a similar model of FGFR-inhibitor-resistant SNU-16 cells [103], and further demonstrated that the induction of EMT was driven by the increased expression of EphB3, as EMT could be reversed, and resistant cells rendered sensitive to EphB3 inhibitor treatment [103].

Likewise, Lau et al. established and cultured a novel *FGFR2*-amplified GC PDX model with the FGFR inhibitor AZD4547 over 14 weeks in vivo [104]. Mechanistically, the authors demonstrated that AZD4547 failed to induce the dephosphorylation of GSK3β in resistant cells, thus preventing its conversion to its active tumour-suppressive state. Notably, GSK3β is a known substrate of PKC and the authors showed that resistance could be overcome by combination treatment of an FGFR inhibitor and the PKC inhibitor H7 [104]. 

Finally, with regards to mechanisms of genetically-driven acquired resistance, gatekeeper mutations that impair drug binding in the ATP-binding pocket of the FGFR kinase domain have been documented in *FGFR2* fusion-positive cholangiocarcinoma [105] and *FGFR3* fusion-positive urothelial cancer patients treated with FGFR inhibitors [106]. While the emergence of these mutations has yet to be clinically documented in FGFR2-driven GCs treated with FGFR inhibitors, a preclinical study did report the emergence of an *FGFR2–V565F* gatekeeper mutation in an *FGFR2*-amplified GC xenograft model continuously treated with an FGFR inhibitor in vitro [104]. 

Other genetic alterations identified in FGFR-inhibitor-resistant GC clones include a *JHDM1D*–*BRAF* fusion, which was observed in a pre-clinical model of acquired resistance [107], while a recent clinical study reported the emergence of an *FGFR2*–*ACSL5* fusion upon progression in a patient with *FGFR2*-amplified GC treated with the FGFR inhibitor LY2874455 [108]. The mechanisms by which these fusion events confer resistance to FGFR inhibitors, and strategies to overcome emergence of these clones, remain to be determined. 

### 5.5. Toxicities Associated with Small-Molecule FGFR Inhibitors

A further limitation of pan-FGFR inhibitors is treatment-associated toxicities. These include hyperphosphatemia, which is observed in >50% of patients in phase II clinical trials and attributed to FGFR1 inhibition; and diarrhea, which occurs in 15–35% of patients and is attributed to FGFR4 inhibition [90]. FGFR2-selective inhibitors including lirafugratinib discussed above, and FGFR2-targeting antibodies discussed below, have the potential to circumvent these toxicities by enabling more targeted FGFR2 inhibition. 

## 6. FGFR2-Targeting Monoclonal Antibodies

An additional class of emerging FGFR2-targeting therapeutics are monoclonal antibodies targeting the extracellular domain of FGFR2-IIIb (FGFR2b), thereby preventing ligand binding. Monoclonal antibodies can be advantageous due their high specificity for molecular targets and potential to induce additional anti-tumour effects such as antibody-dependent cell-mediated toxicity (ADCC) [109], and a number of FGFR2-targeting antibodies (e.g., GP369, GAL-FR21, PRO-007 and bemarituzumab) have shown efficacy in pre-clinical models of *FGFR2*-amplified GC.

Almost a decade ago, GP369 was shown to effectively inhibit FGF7-ligand-induced FGFR2 phosphorylation, MAPK signalling and proliferation of *FGFR2*-amplified SNU-16 GC cells in vitro, and to inhibit their growth in vivo [110]. Likewise, Galaxy Biotech developed a series of FGFR2-targeting mAb’s (GAL-FR21, GAL-FR22 and GAL-FR23), of which GAL-FR21 was shown to selectively bind the FGFR2-IIIb isoform, block the binding and phosphorylation of FGFR2 by its ligands FGF2, FGF7 and FGF10 in SNU-16 cells, and inhibit the growth of *FGFR2*-amplified GC cell lines (SNU-16, OCUM-2) in xenograft models [111]. However, neither of these agents have progressed into the clinic. Similarly, the FGFR2-targeting mAb PRO-007 reduced proliferation, invasiveness and MAPK signalling in *FGFR2*-amplified Kato III GC cells in vitro; however, details regarding its FGFR2 isoform specificity and ability to inhibit FGF-induced FGFR2 stimulation were not provided [112].

More recently, Amgen has developed and commenced clinical testing of bemarituzumab (FPA144), a humanized IgG1 monoclonal antibody targeting FGFR2-IIIb, which has also been glycoengineered (afucosylated) to enhance antibody-dependent cell-mediated cytotoxicity (ADCC) against FGFR2-IIIb-expressing tumour cells. In preclinical studies, bemarituzumab inhibited FGF7-induced FGFR2 phosphorylation and the proliferation of SNU-16 cells in vitro and attenuated growth of the *FGFR2*-amplified GC xenografts OCUM-2M and SNU-16 in vivo. The enhanced ADCC activity of bemarituzumab was also demonstrated by its >20-fold higher affinity for human FcγRIIIa compared to a fucosylated version of the antibody (FPA-144F), and its capacity (but not an ADCC-deficient version, bemaritzumab-N297Q) to suppress tumour growth and increase the recruitment of NK cells into the tumour microenvironment in a syngeneic mammary tumour model. Bemarituzumab also augmented antitumour responses in mammary tumour cells when combined with anti-PD1 antibodies, and in SNU-16 cells when combined with chemotherapy, suggesting it may also enhance the efficacy of existing treatments [113].

Based on these pre-clinical findings, the clinical efficacy of bemarituzumab was initially assessed in a phase I study of pre-treated patients with gastric or GEJ adenocarcinoma. Amongst the 28 patients with *FGFR2* amplification, five patients (17.9%) achieved a partial response and 13 patients (46.4%) had stable disease [114] (Table 3). The follow-up, multicentre, multinational FIGHT trial subsequently randomised 155 patients with FGFR2-IIIb IHC-positive (2+ or 3+ membranous staining in >0% of tumour cells) gastric or GEJ adenocarcinoma to FOLFOX (5-FU, folinic acid, oxaliplatin) chemotherapy plus bemarituzumab or FOLFOX plus placebo. While the trial did not meet its primary endpoint of PFS (HR 0.68 95% CI 0.44–1.04; *p* = 0.073), higher therapeutic efficacy was observed in the bemarituzumab (ORR of 53%) compared to the placebo group (40%) [115]. Furthermore, the median overall survival with prolonged follow up was 19.2 months (95% CI 13.6-NR) and 13.5 months (9.3–15.3) in the bemarituzumab and placebo groups, respectively (HR 0.60 95% CI 0.38–0.94) [116]. Importantly, this trial recruited patients with FGFR2-IIIb-positive IHC expression (30% of screened patients), which is a larger group of patients beyond *FGFR2* amplification. However, a pre-specified subgroup efficacy analysis of this trial recently reported greater efficacy in patients with FGFR2b overexpression (2+/3+ staining) in >10% of tumour cells compared to the overall population [117], consistent with the emerging paradigm that tumours with homogeneous or high-level FGFR2 overexpression are the most likely to benefit from FGFR2-targeted therapies.

### 6.1. FGFR2-Targeting Antibody Drug Conjugates (ADCs) 

A novel area of recent anticancer drug development is antibody-drug conjugates (ADCs), generated by the covalent attachment of a cytotoxic drug to a monoclonal antibody via a chemical linker. In this regard, BAY 1187982 is a novel ADC developed by Bayer, consisting of the fully human FGFR2-IIIb- and FGFR2-IIIc-targeting monoclonal antibody (mAb BAY 1179470), conjugated to the microtubule-disrupting agent auristatin. In preclinical studies, BAY 1187982 displayed efficient internalization and robust anti-tumour activity both in vitro and in vivo in *FGFR2*-amplified GC cell lines [118]. However, despite these preclinical findings, the phase I study of BAY 1187982 produced no clinical responses in 20 patients with FGFR2-positive cancers, including one patient with oesophageal cancer, and two patients with GC. Furthermore, toxicities including thrombocytopenia, proteinuria and corneal epithelial microcysts led to the study being terminated early [119].

More recently, the FGFR2-targeting antibody BAY 1,179,470 has also been developed as a targeted alpha-particle therapy (TAT), which utilizes a monoclonal antibody to deliver an alpha-particle-emitting payload to tumour cells. Specifically, the alpha-particle-emitting radionuclide thorium-227 was conjugated to BAY 1179470 via a chelator moiety to generate an FGFR2-targeted thorium-227 conjugate (FGFR2-TTC, BAY 2304058). Pre-clinical biodistribution studies confirmed the tumour-specific targeting of FGFR2-TTC, which also demonstrated promising pre-clinical efficacy in *FGFR2*-amplified xenograft models including SNU-16 GC cells [120]. The clinical efficacy of this agent remains to be determined. 

### 6.2. Current FGFR2-Targeting Antibody-Based Clinical Trials in GC

The participants in the early phase trials of bemarituzumab may have been highly selected to enrich the trial results for benefit and its efficacy is yet to be confirmed in suitably powered phase III randomised studies. Therefore, bemarituzumab is currently being tested as first-line therapy with chemotherapy or anti-PD1 therapy in two phase III trials. FORTITUDE-101 (NCT05052801) is a double-blind, placebo-controlled phase III trial in patients with untreated, unresectable, locally advanced or metastatic gastric or GEJ adenocarcinoma not amenable to curative therapy, where patients are being randomized to bemarituzumab plus chemotherapy (mFOLFOX6) or placebo plus chemotherapy. Comparatively, in FORTITUDE-102 (NCT05111626), the same patient groups are being randomized to bemarituzumab plus chemotherapy and nivolumab versus chemotherapy and nivolumab (Table 2). These trials are anticipated to complete recruitment in 2025 and 2026, respectively.

## 7. Emerging FGFR2-Targeted Therapies

Several innovative new approaches for targeting FGFR2 are also beginning to emerge in the pre-clinical setting and have been elegantly reviewed recently [101]. These include proteolysis-targeting chimeras (PROTACs), chimeric antigen receptor (CAR)-T cells and soluble receptors. Among these agents, those that have been shown to effectively target FGFR2 in GC models are the FGFR2-targeting PROTACs DGY-09–192 (which also targets FGFR1) [121], and LC-MB12 [122], which comprise the FGFR inhibitor infigratinib linked to recruiters of the E3 ubiquitin ligases VHL or cereblon, respectively. Both compounds effectively degraded FGFR2 in GC cells [121,122], and interestingly, LC-MB12 preferentially degraded FGFR2 over other FGFRs despite infigratinib being a pan-FGFR kinase inhibitor. While the mechanism for this unexpected FGFR2 degrading specificity remains to be determined, LC-MB12 displayed superior inhibition of proliferation and downstream signaling in *FGFR2*-amplified GC cells, compared to infigratinib, in vitro and in vivo. Clinical testing of these agents is now awaited.

## 8. Conclusions

*FGFR2*-amplified GC carries a very poor prognosis and is in urgent need of new treatments. While the recent development of FGFR2-targeting therapies, including small-molecule inhibitors, and FGFR2-specific antibodies have shown activity in pre-clinical models and early phase trials, their clinical efficacy has so far not been sufficient to warrant clinical approval. Nevertheless, small-molecule FGFR2 kinase inhibitors have shown efficacy in GCs with high-level *FGFR2* amplification [94], and more refined trials of these agents in these well-defined molecular subsets may lead to their approval. Pre-clinical studies have also identified several potential drug combinations that can overcome inherent and adaptive resistance to single-agent FGFR inhibitors, which could also be fruitful areas of clinical investigation and identify avenues to clinical approval. The FGFR2-IIIb-targeting antibody bemarituzumab, which inhibits FGFR2 signalling and also induces ADCC, has also shown clinical activity in randomised phase II trials and is now being investigated in combination with chemotherapy and chemotherapy plus immunotherapy in the FORTITUTE trials, the outcomes of which are being eagerly awaited. Novel strategies for FGFR2-targeting also continue to be developed including FGFR2-specific small-molecule inhibitors, antibody-based approaches such as ADCs and TATs, and target-degrading approaches using PROTACs. Overall, given the progress to date and the breadth of preclinical and clinical research currently underway to target this key oncogenic driver in GC, FGFR2 inhibitors are likely to have a meaningful clinical impact in patients with FGFR2-driven GC in the future. 

## Figures and Tables

**Figure 1 biomedicines-12-01117-f001:**
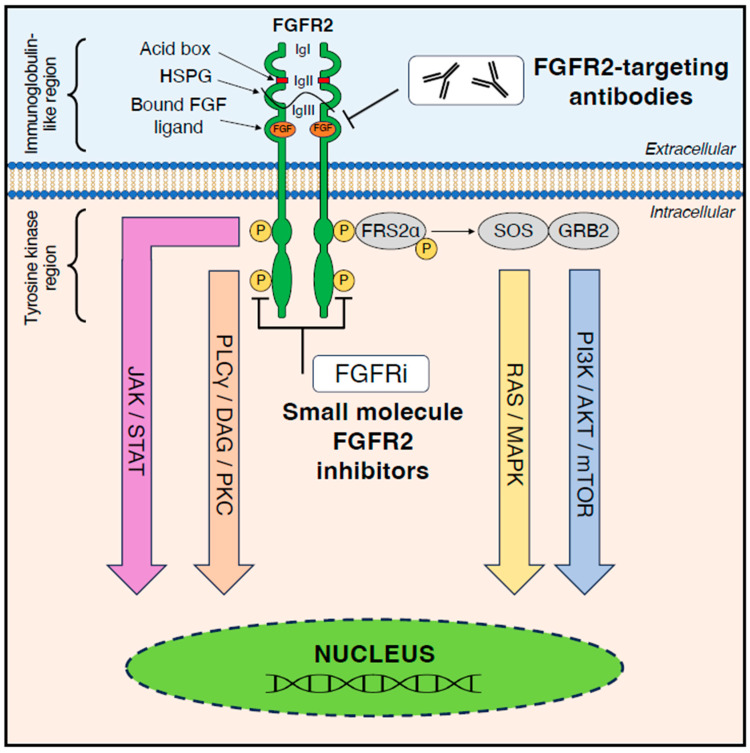
Schematic diagram of the four major FGFR signalling pathways and location of binding sites for FGFR-targeting agents. FGFR 1–4 monomers are comprised of an extracellular region with three immunoglobulin-like domains, a transmembrane domain, and an intracellular region containing two tyrosine kinase domains. FGF binding to FGFR (which is stabilised by HSPG) triggers receptor dimerization and FRS2α phosphorylation. Phosphorylated FRS2α is then able to recruit SOS and GRB2, which initiates the RAS/MAPK and PI3K/AKT/mTOR signalling pathways. Additional signalling pathways initiated by FGFR activation include JAK/STAT and PLCγ/DAG/PKC. FGFR-targeting agents, including small-molecule FGFR inhibitors and FGFR-targeting monoclonal antibodies, bind to the tyrosine kinase domain and the immunoglobulin domain of FGFR, respectively, to inhibit downstream signalling output. Abbreviations: FGF—fibroblast growth factor, FGFR—fibroblast growth factor receptor, FGFRi: fibroblast growth factor receptor inhibitor.

**Table 1 biomedicines-12-01117-t001:** List of FGFR2-targeting therapeutics by mode of action.

Multikinase Inhibitors	Pan-FGFR Inhibitors	FGFR2 Inhibitor(s)	Monoclonal Antibodies	Antibody Drug Conjugates
Cediranib	Erdafitninb	Lirafugratinib	GP369	BAY 1187982
Ponatinib	Infigratinib		GAL-FR21	BAY 1179470
Pazopanib	Pemigatinib		PRO-007	BAY 2304058
Dovitinib	Ch5183284		Bemaritzumab	
Sunitinib	AZD4547		GAL-FR21	
SOMCL-085	LY2874455		GAL-FR22	
Derazantinib	E7090		GAL-FR23	
ODM-203	FIIN-1			
Regorafenib	PRN-1371			
	Futibatinib			
	Alofanib			

**Table 2 biomedicines-12-01117-t002:** Trials in progress of FGFR-targeted therapy in gastro-oesophageal cancer. Abbreviations: mFOLFOX6—5-fluorouracil, folinic acid, oxaliplatin.

Name Clinicaltrials.gov	Phase	Design	FGFR Targeting	Target Recruitment
FIDES-3 NCT04604132	Ib/II	Derazantinib vs. Derazantinib-paclitaxel-ramucirumab vs. Derazantinib-atezolizumab vs. Paclitaxel-ramucirumab	*FGFR2* fusions/rearrangements/amplifications; *FGFR1*, *FGFR2*, or *FGFR3* mutations/short variants	254 (47 actual)
FORTITUDE-102 NCT05111626	Ib/III	Placebo with mFOLFOX6 and Nivolumab vs. Bemarituzumab with mFOLFOX6 and Nivolumab	FGFR2b overexpression	528
FORTITUDE-101 NCT05052801	III	Bemarituzumab with mFOLFOX6 vs. Placebo with mFOLFOX6	FGFR2b overexpression	516
NCT02699606	IIa	Erdafinitib	*FGFR* mutations, translocations, other	90 (actual 35)
NCT05019794	II	Infigratinib	*FGFR2* amplification	80
NCT04189445	II	Futibatinib	*FGFR2* amplification	115 (actual 115)
NCT04526106	I/II	Lirafugratinib (RLY-4008)	*FGFR2* fusion, mutation, or amplification	550

**Table 3 biomedicines-12-01117-t003:** Selected clinical studies of FGFR targeted therapy in gastro-oesophageal cancer. Abbreviations: ORR—objective response rate, PFS—progression free survival, OS—overall survival, mFOLFOX6—5-fluorouracil, folinic acid, oxaliplatin.

Study	Treatment	*n*	ORR	PFS Months	OS Months	Key Eligibility Criteria
Meric-Bernstam [88]	Futibatinib	9	22%	-	-	*FGFR2* amplification
Van Cutsem [89]	AZD4547 Paclitaxel	41 30	2.6% 23.3%	1.8 3.5	5.6 6.6	*FGFR2* amplified and polysomy
Pearson [94]	AZD4547	9	33%	-	-	*FGFR2* amplification
Catenacci [114]	Bemarituzumab	28	17.9%	-	-	FGFR2b IHC High
Wainberg [115]	Bemarituzumab + mFOLFOX6 FOLFOX6	77 78	53% 40%	9.5 7.4 HR 0.68 *p* = 0.07	19.2 13.5 HR 0.60 95% CI 0.38–0.94)	FGFR2b IHC 2+/3+

## Data Availability

No new data were created.

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
