# Peer review of "Clinical Developments and Challenges in Treating FGFR2-Driven Gastric Cancer"

_biomedicines, 2024, doi:10.3390/biomedicines12051117_

Round 1

Reviewer 1 Report (Previous Reviewer 2)

Comments and Suggestions for Authors

The authors have answered my comments and adapted the text and illustrations accordingly.

Author Response

Thank you

Reviewer 2 Report (New Reviewer)

Comments and Suggestions for Authors

The manuscript discusses role of FGFR signaling in gastric cancer and therapeutic benefits of targeting FGFR.  its overall a well written piece but few issues exist which should be addressed by the authors.

1. in line 34, ref no. 2 is not correct citation as its about oesophagal cancer and not GC. also, authors talk about molecular characterization only on genetic basis here. it should also include based on protein biomarkers. for this, a recently published paper should be cited - Li et al., 2023 - A molecular classification of GC....   

2. sections and subsection should be numbered for the readers to guide through the review. 

3. chemical structures of at least some molecules in table 1 should be depicted. 

Comments on the Quality of English Language

English is ok. 

Author Response

The manuscript discusses role of FGFR signaling in gastric cancer and therapeutic benefits of targeting FGFR.  its overall a well written piece but few issues exist which should be addressed by the authors.

  1. in line 34, ref no. 2 is not correct citation as its about oesophagal cancer and not GC. also, authors talk about molecular characterization only on genetic basis here. it should also include based on protein biomarkers. for this, a recently published paper should be cited - Li et al., 2023 - A molecular classification of GC....   

Thank you for picking this up.  We have now added reference PMID:25079317 in this position.  Please note that the original Reference 2 is also correctly cited here, as although this study focused on oesophageal cancer, it also contains a detailed comparison to Gastric cancer.  

We have also included a description of the findings by Li et al, as requested by the Reviewer (PMID: 36700042, as well as other studies which have described methods for classifying gastric cancers based on integrating genomic and non-genomic methods (PMID: 25894828).

  1. Sections and subsection should be numbered for the readers to guide through the review. 

This has now been done, and some sections rearranged to improve the flow, thank you.

  1. Chemical structures of at least some molecules in table 1 should be depicted. 

Thank you for the suggestion. The chemical structures have been well published by the developers of the respective compounds and are also publicly available. We believe the inclusion of chemical structures in this paper would not add improve the quality of the manuscript. As none of the authors are experts in the chemistry of small molecule inhibitors, we would not be able to provide critical commentary on these structures.

Reviewer 3 Report (New Reviewer)

Comments and Suggestions for Authors

The review is well-structured, providing a comprehensive and in-depth analysis of FGFR2 as a target in gastric cancer therapy. It balances molecular biology, clinical developments, and practical challenges, offering a holistic view of the field. I would suggest improving the paper to enhance clarity, depth, and impact. Here are some suggestions:

While the introduction sets the stage well, it could benefit from a brief discussion on the transition to targeted therapies like FGFR2 inhibition. Current introduction did not highlight how FGFR2 targeting represents a new paradigm.

Provide a more detailed examination of the criteria and biomarkers employed to identify candidates for FGFR2-targeted therapies. Details on how these strategies might improve treatment outcomes would be particularly useful.

While the paper discusses resistance to FGFR2-targeted therapies, a deeper exploration on the molecular mechanisms underlying this resistance and potential strategies to overcome it would add value, as reviewed by recent paper on nature reviews clinical oncology (FGFR-targeted therapeutics: clinical activity, mechanisms of resistance and new directions). I would strongly recommend the authors to refine the corresponding section to reflect their adding value on this subject.

 Expanding the discussion on future directions to include emerging research areas, potential novel FGFR2 inbitors such as peptide drugs, and integration with other treatment modalities could provide readers with a sense of where the field is headed.

Comments on the Quality of English Language

The paper demonstrates a high level of English language proficiency, with minor areas for enhancement to improve clarity and depth.

Author Response

The review is well-structured, providing a comprehensive and in-depth analysis of FGFR2 as a target in gastric cancer therapy. It balances molecular biology, clinical developments, and practical challenges, offering a holistic view of the field. I would suggest improving the paper to enhance clarity, depth, and impact. Here are some suggestions:

While the introduction sets the stage well, it could benefit from a brief discussion on the transition to targeted therapies like FGFR2 inhibition. Current introduction did not highlight how FGFR2 targeting represents a new paradigm.

Thank you. We have now expanded the paragraph at the end of the general introduction on Gastric cancer to improve the transition to FGFR2-targeted therapies. We have also added a sentence on page 4 to emphasize that FGFR2 targeting represents a potential new treatment for FGFR2-driven gastric cancers.

Provide a more detailed examination of the criteria and biomarkers employed to identify candidates for FGFR2-targeted therapies. Details on how these strategies might improve treatment outcomes would be particularly useful.

Thank you. We have now added additional detail on potential criteria which can be used to identify patients likely to respond to FGFR2-targeted therapies, including the recent finding that patients with FGFR2b overexpression, determined by IHC staining scores of 2+ or 3+ in >10% of tumour cells respond preferentially to bemarituzumab.

While the paper discusses resistance to FGFR2-targeted therapies, a deeper exploration on the molecular mechanisms underlying this resistance and potential strategies to overcome it would add value, as reviewed by recent paper on nature reviews clinical oncology (FGFR-targeted therapeutics: clinical activity, mechanisms of resistance and new directions). I would strongly recommend the authors to refine the corresponding section to reflect their adding value on this subject.

Thank you. Based on the suggested review, we have now included additional detail on mechanisms of resistance to FGFR2 inhibitors, including TP53 mutation status (page 11).  

Expanding the discussion on future directions to include emerging research areas, potential novel FGFR2 inhibitors such as peptide drugs, and integration with other treatment modalities could provide readers with a sense of where the field is headed.

Thank you. We have now included a new section (section 6) describing emerging new strategies for targeting FGFR2, including the use of PROTACs.

This manuscript is a resubmission of an earlier submission. The following is a list of the peer review reports and author responses from that submission.

Round 1

Reviewer 1 Report

Comments and Suggestions for Authors

The authors are encouraged to address the following suggestions for the improvement of the review article:

  1. Strengthen the impact of recent advances in gastric cancer treatment by incorporating quantitative data that highlights the improvement in survival outcomes.
  2. Emphasize the global significance of gastric cancer mortality by expanding on its context on a worldwide scale. Provide readers with a broader perspective to enhance their understanding of the broader implications.
  3. Elaborate on the current status of chemotherapy as the primary standard therapy for gastric cancer. Include a succinct overview of recent developments in chemotherapy to keep the readers well-informed.
  4. Enhance the depth of the discussion by offering a more detailed explanation of molecular targets beyond FGFR2. Provide readers with a comprehensive overview of the diverse molecular landscape in gastric cancer.
  5. Increase precision in conveying information by specifying a more accurate and concise incidence range for FGFR2 amplification in gastric cancers.
  6. Enrich the discussion by incorporating comparative data on the frequency of FGFR2 amplification relative to other actionable molecular targets in gastric cancer.
  7. Highlight the timeliness of clinical testing for FGFR inhibitors within the framework of recent molecular characterizations. Underline the relevance of these findings to emphasize their immediate impact on advancing gastric cancer treatment.
  8. Explicitly draw attention to existing therapeutic gaps in the treatment of FGFR2-amplified gastric cancers.

9.     Elaborate on the specific parameters used to assess the efficacy of FGFR2-targeting therapies.

10.  Provide references for the statement regarding the warm reception of FGFR2-targeting therapies in the clinic.

11.  Include a brief comparative analysis of different FGFR2-targeting therapies, highlighting their respective strengths and limitations.

12.  Expand on the mechanism by which the FGFR2-IIIb-targeting antibody bemarituzumab induces Antibody-Dependent Cell-Mediated Cytotoxicity (ADCC).

13.  Discuss the anticipated timeline for the potential clinical impact of FGFR2 inhibitors in patients with FGFR2-driven GC.

14.  Discuss key challenges that need to be overcome in the development and approval of FGFR2 inhibitors.

15.  Highlight how the integration of preclinical and clinical research efforts has influenced the progress in targeting FGFR2.

Comments on the Quality of English Language

Moderate editing of English language required

Reviewer 2 Report

Comments and Suggestions for Authors

The present manuscript provides an interesting overview of the action mechanisms of FGFR signaling in gastric cancer as a putative target for novel therapeutic approaches. 

My major comment concerns the clinical data . Recently, doubts have been casted on the quality of clinical trials and the conclusions drawn therefrom. (for example  Guy Storme: Are we losing the final fight against cancer? Cancers 16, 2024, and references therein). The present presentation of clinical data lacks criticality. Terms such as  “a further overall survival benefit” (L42), “more durable responses” and “FDA-approval” (L43), “poorer outcome” (L147), “appeared to benefit” (L189), “begun clinical testing” (L 362), “promising (How promising is an ORR of 53% compared to 40%?) therapeutic efficacy “ (L383), “provided benefit” (L388),  “encouraging results” (L412),  “promising clinical activity” (L433),  “considerable cause for optimism” (L438)  are too vague, give at best false hope. I would suggest a separate section (include all clinical data in section 7?) implicating notes of caution about clinical trials (selection of participants, comparability between groups, number of participants, conflict of interest, confounding control, statistical methods, selective reporting, assessment of the outcome, follow up).

One detail. A graphical presentation (on the basis of Figure 1) or a table  of FGFR inhibitors and their site of action would be useful.